# Prevalence and clinical implications of respiratory viruses in stable chronic obstructive pulmonary disease (COPD) and exacerbations: a systematic review and meta-analysis protocol

Anastasia M Kefala,[1] Rebecca Fortescue,[2] Gioulinta S Alimani,[1,3] Prodromos Kanavidis,[4] Melissa Jane McDonnell,[5] Emmanouil Magiorkinis,[1,6] Spyridon Megremis,[7] Dimitrios Paraskevis,[8] Chrysa Voyiatzaki,[1] Georgios A Mathioudakis,[3] Effie Papageorgiou,[1] Nikolaos G Papadopoulos,[9,10] Jørgen Vestbo,[9,11] Apostolos Beloukas [ID],[1,12] Alexander G Mathioudakis [ID] [9,11]

For numbered affiliations see end of article.

**Correspondence to**
Dr Alexander G Mathioudakis; alexander.mathioudakis@manchester.ac.uk

## ABSTRACT

**Introduction** Both stable chronic obstructive pulmonary disease (COPD) and acute exacerbations represent leading causes of death, disability and healthcare expenditure. They are complex, heterogeneous and their mechanisms are poorly understood. The role of respiratory viruses has been studied extensively but is still not adequately addressed clinically. Through a rigorous evidence update, we aim to define the prevalence and clinical burden of the different respiratory viruses in stable COPD and exacerbations, and to investigate whether viral load of usual respiratory viruses could be used for diagnosis of exacerbations triggered by viruses, which are currently not diagnosed or treated aetiologically.

**Methods and analysis** Based on a prospectively registered protocol, we will systematically review the literature using standard methods recommended by the Cochrane Collaboration and the Grading of Recommendations Assessment, Development and Evaluation working group. We will search Medline/PubMed, Excerpta Medica dataBASE (EMBASE), the Cochrane Library, the WHO's Clinical Trials Registry and the proceedings of relevant international conferences on 2 March 2020. We will evaluate: (A) the prevalence of respiratory viruses in stable COPD and exacerbations, (B) differences in the viral loads of respiratory viruses in stable COPD vs exacerbations, to explore whether the viral load of prevalent respiratory viruses could be used as a diagnostic biomarker for exacerbations triggered by viruses and (C) the association between the presence of respiratory viruses and clinical outcomes in stable COPD and in exacerbations.

**Ethics and dissemination** Ethics approval is not required since no primary data will be collected. Our findings will be presented in national and international scientific conferences and will be published in peer reviewed journals. Respiratory viruses currently represent a lost opportunity to improve the outcomes of both stable COPD and exacerbations. Our work aspires to 'demystify' the prevalence and clinical burden of viruses in stable COPD

## Strengths and limitations of this study

► A holistic approach to a clinically pertinent question that could pump-prime clinical and translational research and facilitate the introduction of precision medicine interventions both for stable obstructive pulmonary disease and exacerbations.

► A rigorous methodology that includes a thorough evaluation of the literature, appropriate evaluation of the risk of bias of individual studies and the quality of the body of evidence, and quantitative synthesis.

► Several planned meta-regression analyses to explore potential effect modifying factors; however, a potential limitation is data may not be available for the completion of all these analyses.

► A prospectively published protocol increases the transparency and allowed for peer review of the methodology used.

and exacerbations and to promote clinical and translational research.

**PROSPERO registration number** CRD42019147658.

## INTRODUCTION

Chronic obstructive pulmonary disease (COPD), a leading cause of death, disability and health-related expenditure globally, is characterised by chronic debilitating respiratory symptoms and acute exacerbations, that drive the adverse outcomes of the disease.[1–4] Both stable COPD and exacerbations are complex and heterogeneous, meaning that several distinct mechanistic pathways contribute to their development and progression.[5 6] While some of these pathways are well established, our understanding of the pivotal

role that respiratory viruses appear to have both in stable disease and exacerbations is still lacking.

Viruses are identified in over 10% of all patients with stable COPD at any given time[6 7 8] and in 30%–50% of those experiencing exacerbations.[9–11] In both cases, the presence of viruses is associated with worse clinical outcomes.[6 7 8] For example, the prevalence and clinical implications of viruses were tested longitudinally in a subgroup of 127 patients with COPD, who experienced 355 exacerbations during 1 year of follow-up from the Acute Exacerbation and Respiratory InfectionS in COPD (AERIS) cohort[8] and 83 patients, who experienced 168 exacerbations during follow-up from the East London COPD cohort.[7] In the first cohort, viruses were present in 13.6% and 41.3% of samples received during stable disease and exacerbations, respectively. Almost half of the participants (46.5%) had at least one exacerbation that tested positive for human rhinovirus (HRV) during 1 year of follow-up. In the East London COPD cohort, viruses were detected in 39.2% of all exacerbations and were associated with higher symptoms burden and delayed recovery. The presence of viruses during the stable state was associated with more frequent exacerbations.[7] In general, rhinovirus, influenza and respiratory syncytial virus are the most frequently detected and their clinical burden has been demonstrated in several studies.[7–9 11]

Diagnostic techniques for exacerbations triggered by respiratory viruses are currently inadequate. Respiratory viruses are prevalent both in exacerbations and stable COPD. Therefore, the presence of viruses in the airways cannot confirm viral aetiology of an exacerbation, since viruses may represent incidental findings, rather than the real cause of the exacerbation.

Acute respiratory viral infections in healthy people are characterised by quick viral replication leading to high viral shedding and load that peak rapidly.[12] Assuming that the same pattern is followed in patients with COPD, we would anticipate higher viral loads in exacerbations triggered by viruses (not only compared with stable COPD, but also compared with exacerbations in which viruses are present but not the causative factor). Indeed, previous studies have revealed 10-fold higher mean viral loads of HRV in exacerbations characterised by symptoms consistent with a respiratory viral infection compared with stable COPD,[10 13] suggesting that viral load might be used as a diagnostic biomarker for exacerbations triggered by viruses. Accurate diagnosis of acute exacerbations triggered by viruses could lead to the introduction of targeted antiviral treatments that are already commercially available or in development.[14] Additionally, it could lead to a reduction of the unnecessary administration of antibiotics for exacerbations that are triggered by viruses (and not by bacteria).[15 16]

There is an astonishing imbalance between the ample published evidence on the prevalence and clinical impact of respiratory viruses in COPD and the lack of insight on their clinical approach and management. To improve our understanding and facilitate the introduction of precision medicine interventions for this group of patients, we will conduct a systematic review aiming to holistically evaluate:

A. the prevalence of respiratory viruses in patients with stable COPD and exacerbations.
B. differences in the viral loads of respiratory viruses in stable COPD vs exacerbations, to explore whether viral loads of prevalent respiratory viruses could be used as a gold standard for the diagnosis of exacerbations triggered by those viruses.
C. the association between the presence of respiratory viruses and clinical outcomes of patients with stable COPD and exacerbations.

## METHODS AND ANALYSIS

This systematic review and meta-analysis will be based on a prospectively registered protocol at the International Prospective Register of Systematic Reviews (PROSPERO) and it will be conducted following standard methodology recommended by the Cochrane Collaboration[17] and the Grading of Recommendations Assessment, Development, and Evaluation (GRADE) working group.[18] The Preferred Reporting Items for Systematic Reviews and Meta-Analysis statement was used for reporting this protocol[19] and will also be followed for the preparation of the final report of this systematic review.[20]

### Eligibility criteria

We will include studies evaluating patients aged over 40 years with COPD, either during stable disease state or exacerbations. We will consider eligible studies evaluating stable COPD, provided that participants had a clinical diagnosis of COPD, confirmed by spirometry. For studies evaluating patients during exacerbations, a previous clinical diagnosis of COPD will suffice. Spirometry during an exacerbation has poor specificity, and for this reason, it will not be a prerequisite for studies on exacerbations.[21 22] We will exclude studies focusing on specific populations who are at high risk of viral infections, such as lung transplant recipients, or people with immune deficiency.

Only studies using molecular techniques to identify respiratory viruses will be included. For projects A and B (evaluating the prevalence and loads of respiratory viruses), we will include studies exploring the prevalence of respiratory viruses in representative samples of patients with stable COPD or exacerbations. Only studies assessing associations between the presence and/or load of respiratory viruses and clinical outcomes of COPD or exacerbations will be included in project C.

We will include studies of any design that could contribute data pertinent to the review questions. Data from randomised controlled trials (RCTs) will be used more cautiously compared with observational studies. More specifically, we will only include data that are unlikely to be affected by investigational medications that are not part of the usual care for COPD or exacerbations. For example, in a trial evaluating the impact of a novel immunomodulatory agent versus usual care on

exacerbations triggered by viruses, we will only capture data on the exacerbations' outcomes from the control group. We would capture data on the prevalence and loads of respiratory viruses from both study groups, provided they were measured prior to the initiation of antiviral treatments, in the intervention group.

## Outcome measures

Projects A and B: primary outcome: prevalence of respiratory viruses in the respiratory tract of patients with stable COPD and exacerbations. We will assess individual viruses separately, but also the presence of any virus. Secondary outcomes: mean viral loads of respiratory viruses during stable COPD and exacerbations; seasonal variability in the prevalence and viral loads of respiratory viruses and proportion of patients testing positive for more than one virus. We will also describe the molecular techniques and laboratory tests used for identification and quantification of respiratory viruses.

For studies evaluating stable COPD, the main analysis will only include the first assessment of respiratory viruses in every patient. In case of multiple measurements during different seasons of the year, we will capture the first measurement during every season, to be included in the analysis of seasonal variability. For studies evaluating exacerbations, the main analysis will include the first assessment of respiratory viruses during every exacerbation. The unit of analysis will be the exacerbation, rather than the patient. In a sensitivity analysis, we will only include the first exacerbation captured for every participant.

Project C: co-primary outcomes for studies evaluating stable COPD: annual rate of moderate or severe exacerbations and mortality. Secondary outcomes: annual rate of severe exacerbations; symptoms severity; health-related quality of life; exercise capacity; forced expiratory volume in 1 s ($FEV_1$) decline rate.

Co-primary outcomes for studies evaluating exacerbations: mortality and treatment failure rate. Secondary outcomes: treatment success rate; symptoms severity; symptoms duration; length of hospitalisation; time-to-next exacerbation; exacerbations frequency; co-existing bacterial infection and proportion of patients who received antiviral treatment.

Treatment failure in exacerbations is usually defined as a composite outcome, including several of the following components: lack of clinical improvement, symptoms deterioration, hospital admission, intensive care unit admission, need for additional treatments or death.[23] Treatment success is defined as complete resolution or significant improvement of the symptoms.[23]

All clinical outcomes will primarily be evaluated at longest follow-up, with the exception of treatment failure and treatment success rates in studies evaluating exacerbations that will be evaluated at 1–4 weeks from presentation. In additional analyses of stable COPD studies, we will evaluate separately data collected between 0– <3, 3– <6, 6– <12 and ≥12 months of follow-up. For exacerbations studies, in additional analyses, we will assess separately data collected between 0– <1, 1– <2, 2– <6 and ≥6 months of follow-up, to better capture the early outcomes of exacerbations.

## Systematic literature search

The electronic databases of Medline/PubMed, Excerpta Medica dataBASE (EMBASE) and the Cochrane Central will be searched on 2 March 2020, using a comprehensive search strategy including appropriate controlled vocabulary and free search terms. This strategy was developed by one author (AGM) and was updated following input from all authors and identification of Medical Subject Headings (MeSH) terms from several eligible studies that were identified during pilot searches. In addition to the online databases, we will search for relevant studies in the conference proceedings of the European Respiratory Society, American Thoracic Society and Asian Pacific Society of Respirology, European Society of Clinical Microbiology and Infectious Diseases, American Society of Microbiology, European Society for Clinical Virology, in the WHO International Clinical Registry Platform and in the reference lists of all included studies and all previously published systematic reviews. Detailed search strategies are available in the online supplementary appendix.

Two investigators will independently screen the titles and abstracts of all studies that our searches will yield. Next, the full-text versions of all potentially eligible manuscripts and abstracts will be acquired and reviewed for confirmation of eligibility by two authors independently. Disagreement in this and all following steps of the systematic review process will be resolved by discussion or adjudication by a third investigator, when necessary.

## Data abstraction

Relevant data from each eligible study will be extracted in a predefined, pilot-tested excel spreadsheet. The full reference of each eligible study, as well as details on the study design, eligibility criteria, baseline characteristics, details on the viruses evaluated and the performance characteristics of laboratory assays used for viral identification and/or quantification will be extracted by one person and will be cross-checked by another person for accuracy. Data regarding the prevalence and viral loads of respiratory viruses and all clinical outcomes will be extracted by two investigators independently. Details on all data and variables that will be extracted are available in the online supplementary appendix.

Pilot testing of the data extraction form on a sample of studies will be conducted by all investigators who will subsequently use the form. In case of discrepancies, additional eligible studies will be extracted until achieving an agreement in >95% of the extracted variables among all reviewers. Feedback will be sought during this process and variables may be added or modified. Missing data will be requested directly from the study investigators via email.

### Risk of bias assessment

For projects A and B, which focus on the prevalence and viral loads of viruses in the respiratory tract of patients with stable COPD or exacerbations, we will use the risk of bias tool for prevalence studies developed by Hoy *et al*.[24] For project C, we will use the Newcastle-Ottawa Scale.[25] Both tools were developed for assessing the quality of non-randomised studies. For this reason, they assess more thoroughly the representativeness of the population of the included studies.[24 25] On the contrary, tools evaluating the risk of bias in RCTs focus less on representativeness.[17] However, representativeness of the participants is of paramount importance in our meta-analysis, which does not focus on the trial interventions. For these reasons, we decided to use risk of bias tools for non-randomised studies for risk of bias evaluation in studies of any design, including RCTs. Risk of bias will be assessed by two investigators independently. For each study, we will report an overall risk of bias assessment, but also our judgement of the risk of bias related to each of the domains proposed by the selected tools. If we can pool >10 studies, we will be able to explore possible small study and publication biases by creating and examining a funnel plot and by using Egger's regression and Begg's rank correlation.[17]

### Data synthesis

Heterogeneity among the studies in each meta-analysis will be estimated using the $I^2$ statistic. Substantial heterogeneity ($I^2 \geq 50\%$) will be reported and possible causes will be explored by prespecified meta-regression analyses.

In the primary analyses, data will be synthesised using a random-effects model because we anticipate significant clinical and methodological heterogeneity among the included studies. Fixed-effects models will be used in sensitivity analyses. The inverse variance method using logit transformed proportions will be used for conducting meta-analysis of proportions. In the main analyses, individual viruses will be evaluated separately. In an additional analysis, we will evaluate the presence of any virus. Meta-analyses will be performed using R statistical software (R Foundation for Statistical Computing, Vienna, Austria) and the CRAN packages for meta-analysis (meta and metafor).

### Meta-regression and sensitivity analyses

If adequate data are available, we will conduct meta-regression analyses to evaluate the potential impact of the following parameters on the outcomes: (1) the season of the year when samples were collected; (2) use of inhaled corticosteroids; (3) spirometric severity of COPD, classified based on their $FEV_1$ ($\geq 80\%$, 50%–80%, 30%–50% or <30%); (4) year of study completion and (5) study design (observational vs interventional). Additional parameters to be tested in project C will include: (6) bacterial co-infection and (7) receipt of antiviral treatment. For each of these analyses, we will first attempt to use subgroup data from the included studies, if they are available to us. Alternatively, we will use the following study-level

variables for each of the analyses: (1) the proportion of samples that were collected during the influenza season; (2) the proportion of patients that were using inhaled corticosteroids; (3) the mean $FEV_1$ of the participants; (4)–(5) same as above; (6) proportion of patients with bacterial co-infection and (7) proportion of patients that received antiviral treatment. For conducting these analyses, we will use a random-effects model and for meta-regression analyses of proportions, we will use the inverse variance method with logit transformed proportions. We will report the level of significance of the regressions (p value) and the correlation coefficients. A risk of type I error is introduced since we plan to evaluate the impact of seven parameters on the outcomes. To mitigate this risk, we will only conduct univariate meta-regression analysis. Moreover, we will only confidently assume effect modification in cases where the p value is <0.01 and we will only consider the possibility of effect modification if p<0.05.

Moreover, we plan to conduct the following sensitivity analyses:

a. For analyses focusing on stable COPD, we will only include studies which only included participants who did not have any exacerbations and had not received any oral corticosteroids for at least 3 months prior to sampling.
b. For analyses focusing on exacerbations, we will only include studies which only included patients with a clinical diagnosis of COPD confirmed by spirometry during stable state.
c. For analyses focusing on exacerbations, we will only include studies providing data for only one exacerbation per participant.
d. We will repeat meta-analyses using fixed-effects models.
e. We will only include studies of low risk of bias.

### Certainty of evidence

For each outcome, we will evaluate the quality of the body of evidence using GRADE methodology, which takes into consideration study limitations, consistency of the effect, imprecision, indirectness, publication bias, magnitude of effect, dose response and confounders likely minimising the effect.[18]

### Protocol deviations

We will conduct our systematic review according to this published protocol. Any deviations will be documented and justified in our final report.

### Patient and public involvement

This systematic review protocol was developed without patient or public involvement.

### Ethics and dissemination

Ethical approval is not a requirement since no primary data will be collected.

The findings of this systematic review and meta-analysis will be presented in national and international scientific conferences. They will also be submitted for publication in high-impact peer-reviewed journals. It is anticipated

## DISCUSSION

Respiratory viral infections represent a major unmet treatment need in both stable COPD and exacerbations. The planned meta-analyses will systemise the existing evidence on their prevalence, clinical burden and—in the case of exacerbations—diagnostic techniques.

The prevalence of viral infections in patients with COPD exacerbations has been evaluated in two previous systematic reviews.[26 27] Both had limitations. Only PubMed was searched, and the most recent literature search was conducted in March 2017. When comparing the studies included in each of these reviews, we can see that important studies have been missed. Moreover, there is no formal assessment of the risk of bias of the included studies. Finally, quantitative analyses did not evaluate the prevalence of specific viruses, but rather the presence of any virus. To the best of our knowledge, the prevalence of viral infections in patients with stable COPD and the clinical implications of the presence of viruses in either stable COPD or exacerbations has not been previously assessed in a systematic review.

Main strengths of our systematic review include the clinically pertinent review questions, thorough searches of the available literature and methodological rigour. The inclusion of several meta-regression and sensitivity analyses will improve our confidence on the certainty of our findings. Potential unavoidable limitations include the lack of data availability on less prevalent viruses (such as metapneumovirus, which may not be assessed in many of the available studies) and technical limitations of the available studies, such as lack of standardisation of the molecular techniques used to identify and quantify respiratory viruses. To address technical heterogeneity, we will report on the performance characteristics of the molecular techniques used. Challenges are anticipated in pooling viral loads, as several studies use relative quantification processes with little or no normalisation. In the main analysis, evaluating the mean viral loads during stable COPD and exacerbations, only studies reporting viral copies will be included. In an additional analysis, evaluating the relation of viral loads between stable COPD and exacerbations, we will also include studies using the same relative quantification process for both disease states.

We expect that this work could stimulate clinical and translational research, facilitate the introduction of precision medicine interventions both for stable COPD and exacerbations and improve the outcomes of these patients.

**Author affiliations**
[1]Department of Biomedical Sciences, University of West Attica, Egaleo, Greece
[2]Cochrane Airways, Population Health Research Institute, University of London Saint George's, London, UK
[3]Athens Breath Centre, Athens, Greece
[4]First Department of Surgery, Laikon General Hospital, National and Kapodistrian University of Athens, Athens, Greece
[5]Department of Respiratory Medicine, Galway University Hospitals, Galway, Ireland
[6]Department of Laboratory Haematology, Sotiria Regional Chest Disease Hospital of Athens, Athens, Greece
[7]Division of Evolution and Genomic Science, The University of Manchester, Manchester, UK
[8]Department of Hygiene, Epidemiology and Medical Statistics, National and Kapodistrian University of Athens, Athens, Greece
[9]Division of Infection, Immunity and Respiratory Medicine, The University of Manchester, Manchester, UK
[10]Allergy Department, 2nd Paediatric Clinic, National and Kapodistrian University of Athens, Athens, Greece
[11]North West Lung Centre, Manchester University NHS Foundation Trust, Manchester Academic Health Science Centre, Manchester, UK
[12]Institute of Infection and Global Health, University of Liverpool, Liverpool, UK

**Contributors** JV, AB and AGM contributed to conception of the study and are the guarantors of this review. AMK, RF, PK, JV, AB and AGM contributed to study design. RF and AGM provided methodological input. AMK, AB and AGM drafted the manuscript. All authors (AMK, RF, GSA, PK, MJM, EM, SM, DP, CV, GAM, EP, NGP, JV, AB and AGM) contributed to critical revision of the manuscript for intellectual content. AB and AGM equally shared last authorship to this work.

**Funding** JV and AGM were supported by the National Institute of Health Research Manchester Biomedical Research Centre (NIHR Manchester BRC).

**Competing interests** DP reports grants form Gilead Sciences, GSK, Janssen and MSD, outside the submitted work. NGP reports personal fees from Novartis, Nutricia, HAL, Menarini/Faes Farma, Sanofi, Mylan/Meda, Biomay, AstraZeneca, GSK, MSD, Asit Biotech, Boehringer Ingelheim and grans from Gerolymatos International SA and Capricare, outside the submitted work. JV reports personal fees from Chiesi Pharmaceuticals, Boehringer Ingelheim, Novartis, AstraZeneca, GSK, outside the submitted work. AGM reports grants from Boehringer Ingelheim, outside the submitted work.

**Patient and public involvement** Patients and/or the public were not involved in the design, or conduct, or reporting, or dissemination plans of this research.

**Patient consent for publication** Not required.

**Provenance and peer review** Not commissioned; externally peer reviewed.

**ORCID iDs**
Apostolos Beloukas http://orcid.org/0000-0001-5639-0528
Alexander G Mathioudakis http://orcid.org/0000-0002-4675-9616

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
