## [Reviewer comments · BMJ Open]

ARTICLE DETAILS

TITLE (PROVISIONAL)	Prevalence and clinical implications of respiratory viruses in stable chronic obstructive pulmonary disease (COPD) and exacerbations: A systematic review and meta-analysis protocol.
AUTHORS	Kefala, Anastasia; Fortescue, Rebecca; Alimani, Gioulinta; Kanavidis, Prodromos; McDonnell, Melissa Jane; Magiorkinis, Emmanouil; Megremis, Spyridon; Paraskevis, Dimitrios; Voyiatzaki, Chrysa; Mathioudakis, Georgios; Papageorgiou, Effie; Papadopoulos, Nikolaos; Vestbo, Joergen; Beloukas, Apostolos; Mathioudakis, Alexander

VERSION 1 - REVIEW

REVIEWER	Alicia Mitchell University of Sydney Medical School Sydney, Australia
REVIEW RETURNED	24-Nov-2019

GENERAL COMMENTS	This protocol is well written and clearly explained. It addresses a very important and relevant clinical question, and concisely describes a well thought out methodology for approaching this systematic review and meta-analysis. Clear guidelines are needed for future approaches to similar clinical questions. I am very interested to see the results of this review when completed, as this is a very topical area of research and data delineating viral and bacterial exacerbations, and varying ways to treat each, are very much needed.
--

REVIEWER	Shawn Aaron The Ottawa Hospital Research Institute
REVIEW RETURNED	28-Nov-2019

GENERAL COMMENTS	This is a protocol for a systematic review. If BMJ Open is interested in publishing protocols for systematic reviews, then this protocol is publishable. I have no suggestions for changes.
---

REVIEWER	Amalia Karahalios The University of Melbourne, Australia
REVIEW RETURNED	23-Dec-2019

GENERAL COMMENTS

This is a protocol for a systematic review that aims to estimate the prevalence and clinical burden of the different respiratory viruses in stable COPD and exacerbations and to investigate whether viral load of usual respiratory viruses could be used for diagnosis of exacerbations triggered by viruses, which are currently not diagnosed or treated aetiologically.

As requested, I have focused and assessed the appropriateness of the methods described to answer the research questions. There are some details of the methods that are lacking and I have raised these below. These are followed by minor comments and suggested revisions.

Major comments:

1. The authors claim that one of the strengths of this study is that there are “Several planned subgroup analyses to account for potential confounding”. It’s not clear how this is reflected in the analysis section.

2. For the search strategies presented in the online/supplementary material:

a. Was a librarian/information specialist consulted in the development of the search strategy?

b. It seems odd that the full text of search term #5 (COPD) is included in the search strategy but not #6 (COAD). Similar for search terms #12 and #13. Please explain.

3. The authors indicate that they will search for conference proceedings in several databases but they don’t explain how they will handle conference proceedings that are relevant but are lacking the necessary information/data to be included in this review.

4. Data extraction – a list of the data that will be extracted should be included as a supplementary file. Further details of the data extraction process are needed including where will the data be extracted, and if there will be any pilot testing of the data extraction forms.

5. The authors indicate that they will use the “Standard methods recommended by the Cochrane Collaboration” and then proceed to state that they will use “the risk of bias tool for prevalence studies developed by Hoy and colleagues²¹. For project C, we will use the Newcastle-Ottawa Scale²².” These tools are not recommended by the Cochrane Collaboration and I have reservations about these tools. Summary scales such as the two that have been proposed involve inherent weight of component items (Greenland 1994 “Quality scores are useless and potentially misleading” American Journal of Epidemiology, 300-301). The Risk Of Bias In Nonrandomised Studies of Interventions (ROBINS-I) tool (Sterne et al. BMJ. 2016) is the tool recommended by Cochrane (see chapter 25 of the Cochrane Handbook version 6, 2019) to assess the risk of bias in non-randomised studies.

Greenland. "Quality scores are useless and potentially misleading." American Journal of Epidemiology. 1994; pages 300-301.

Sterne et al. “ROBINS-I: a tool for assessing risk of bias in non-randomised studies of interventions”. BMJ. 2016;355:i4919.

6. The authors need to provide more detail in the data synthesis section. Specifically, the authors need to consider how they will synthesise data for estimates of prevalence, what

	transformation they will use and if necessary/appropriate what estimator for the between-study heterogeneity will be used. 7. What will the authors do if there aren't enough studies to undertake a meta-analysis? And how will you determine if there are an adequate number of studies? 8. The authors have proposed a total of 10 subgroup and sensitivity analyses. Potentially, the authors will end up with more subgroup/sensitivity analyses than included studies. For this reason, the authors should define a hierarchy of subgroup/sensitivity analyses. 9. As well, in the subgroup and sensitivity analyses the authors describe these analyses in terms of patients (e.g. "We will analyse separately patients who were receiving/not receiving inhaled corticosteroids"), however the authors won't necessarily have the data at the patient level but instead they will have the aggregate data presented in each study. How will the authors' handle the situations where they can not disentangle the different subgroups of patients in the studies? 10. In the discussion section the authors mention 2 previous systematic reviews but they don't cite the references and so it is impossible to know how this review differs from the previous publications and whether this is in fact adding anything more to the literature. Minor comments:  1. Abstract – I don't understand the first sentence of the abstract. 2. Methods and analysis - change PRISMA-p to PRISMA-P (capitalize the final P) and include the appropriate reference. 3. Eligibility criteria – lines 166-173 this section/explanation should be included in the methods rather than the eligibility criteria 4. Change the subheading 'Literature searchers' to "Systematic literature search" 5. Last paragraph of the discussion the author's state "We expect that this work could pump-prime clinical and translational research" – what does 'pump-prime' mean? Please revise. 6. Is the author list for references 1 and 2 correct?
--	--

VERSION 1 – AUTHOR RESPONSE

Reviewer(s)' Comments to Author:

Reviewer: 1

Reviewer Name

Alicia Mitchell

Institution and Country

University of Sydney Medical School
Sydney, Australia

Please state any competing interests or state 'None declared':
None declared

Please leave your comments for the authors below

This protocol is well written and clearly explained. It addresses a very important and relevant clinical question, and concisely describes a well thought out methodology for approaching this systematic review and meta-analysis. Clear guidelines are needed for future approaches to similar clinical questions. I am very interested to see the results of this review when completed, as this is a very topical area of research and data delineating viral and bacterial exacerbations, and varying ways to treat each, are very much needed.

Response: Thank you very much for your interest and kind words. We will share our findings once we finalise our systematic review and meta-analysis.

Reviewer: 2

Reviewer Name

Shawn Aaron

Institution and Country

The Ottawa Hospital Research Institute

Please state any competing interests or state 'None declared':
none

Please leave your comments for the authors below

This is a protocol for a systematic review. If BMJ Open is interested in publishing protocols for systematic reviews, then this protocol is publishable. I have no suggestions for changes.

Response: Thank you very much for your peer review. We consider prospective publication of systematic review protocols important, as it improves transparency and reduces the potential for bias.

Reviewer: 3

Reviewer Name

Amalia Karahalios

Institution and Country

The University of Melbourne, Australia

Please state any competing interests or state 'None declared':
None declared

Please leave your comments for the authors below

This is a protocol for a systematic review that aims to estimate the prevalence and clinical burden of the different respiratory viruses in stable COPD and exacerbations and to investigate whether viral load of usual respiratory viruses could be used for diagnosis of exacerbations triggered by viruses, which are currently not diagnosed or treated aetiologically.

As requested, I have focused and assessed the appropriateness of the methods described to answer the research questions. There are some details of the methods that are lacking and I have raised these below. These are followed by minor comments and suggested revisions.

Major comments:

1. The authors claim that one of the strengths of this study is that there are “Several planned subgroup analyses to account for potential confounding”. It’s not clear how this is reflected in the analysis section.

Response: We thank the reviewer for all her productive comments and suggestions. We have now revised our planned analyses. Firstly, we decided to conduct meta-regression, instead of subgroup analyses, as (i) some of the variables that we intend to evaluate are continuous (e.g. the proportion of participants who were receiving ICS) and (ii) meta-regression analyses allow for a better visualisation of the findings in bubble plots. The sentence pointed out by the reviewer has now been revised: “Several planned meta-regression analyses to explore potential effect modifying factors; however, a potential limitation is data may not be available for the completion of all these analyses.”

In addition, we provide a detailed description of the methodology that will be used in the conduct of the meta-regression analyses (lines 273-290).

We will update the PROSPERO study registration to match the current version of this manuscript.

2. For the search strategies presented in the online/supplementary material: a. Was a librarian/information specialist consulted in the development of the search strategy?
b. It seems odd that the full text of search term #5 (COPD) is included in the search strategy but not #6 (COAD). Similar for search terms #12 and #13. Please explain.

Response: This search strategy has not been informed by a librarian. However, two of the authors have extensive experience in the development of search strategies (Rebecca Fortescue, who is a Co-ordinating Editor in Cochrane Airways, and Alexander Mathioudakis, who is also an Editor in Cochrane Airways). This search strategy was developed by one author (AGM) and was updated following input from all the authors and identification of MeSH terms from several eligible studies that were identified during pilot searches. We have now clarified that in the manuscript: “The electronic databases of Medline/Pubmed, EMBASE and the Cochrane Central will be searched using a comprehensive search strategy using appropriate controlled vocabulary and free search terms. This strategy was developed by one author (AGM) and was updated following input from all authors and identification of MeSH terms from several eligible studies that were identified during pilot searches.”

The full text of search term COAD is included in the strategy: The terms “Chronic Obstructive Pulmonary Disease” and “Chronic Obstructive Airway Disease” are synonyms in the MeSH library, therefore, both terms are covered in #1. Moreover, “Obstructive” and “Airways” are covered as free text terms in lines #9 - #11.

Lines #12 and #13 refer to “Acute exacerbations of chronic obstructive pulmonary disease” and “Infective exacerbations of chronic respiratory disease” and the full text is also covered in lines #1 and #9-#11 (by the more general term chronic obstructive pulmonary disease).

3. The authors indicate that they will search for conference proceedings in several databases but they don’t explain how they will handle conference proceedings that are relevant but are lacking the necessary information/data to be included in this review.

Response: We have now clarified in the data abstraction section that for all included studies (both those described in conference proceedings or full reports), we will contact authors to seek information that may be missing. "Missing data will be requested directly from the study investigators, via e-mail."

4. Data extraction – a list of the data that will be extracted should be included as a supplementary file. Further details of the data extraction process are needed including where will the data be extracted, and if there will be any pilot testing of the data extraction forms.

Response: Data extraction process and piloting of the data extraction form is now described in detail (lines 232-244). In addition, a list of all variables that will be captured in the data extraction form is included in the online appendix.

5. The authors indicate that they will use the "Standard methods recommended by the Cochrane Collaboration" and then proceed to state that they will use "the risk of bias tool for prevalence studies developed by Hoy and colleagues²¹. For project C, we will use the Newcastle-Ottawa Scale²²." These tools are not recommended by the Cochrane Collaboration and I have reservations about these tools. Summary scales such as the two that have been proposed involve inherent weight of component items (Greenland 1994 "Quality scores are useless and potentially misleading" *American Journal of Epidemiology*, 300-301). The Risk Of Bias In Non-randomised Studies of Interventions (ROBINS-I) tool (Sterne et al. *BMJ*. 2016) is the tool recommended by Cochrane (see chapter 25 of the *Cochrane Handbook* version 6, 2019) to assess the risk of bias in non-randomised studies.

Greenland. "Quality scores are useless and potentially misleading." *American Journal of Epidemiology*. 1994; pages 300-301.

Sterne et al. "ROBINS-I: a tool for assessing risk of bias in non-randomised studies of interventions". *BMJ*. 2016;355:i4919.

Response: We thank the reviewer for this comment, but we respectfully disagree. Unfortunately, the ROBINS-I tool is not appropriate for our systematic review, as it is developed to evaluate the risk of bias in non-randomized studies of interventions. In our meta-analysis, we do not evaluate an intervention, but the prevalence of a trait (respiratory viruses), or the impact of respiratory viruses on the clinical outcomes. Therefore, many of the domains evaluated in ROBINS-I tool are not applicable in our meta-analysis and use of this tool might lead to significant misclassification of the methodological quality of studies. Instead we chose to use the scale developed by Hoy and colleagues to evaluate risk of bias in meta-analyses of prevalence and the NOS scale, aimed to evaluate the quality of nonrandomised studies (not necessarily interventional). Therefore, these tools are more specific to our analyses and we believe they will facilitate a more robust evaluation of the methodological quality. However, we agree that reporting a summary score provides inadequate information on the studies' risk of bias and for this reason, we will report our judgement of the risk of bias for each of the domains included in the proposed tools (lines 255-257).

6. The authors need to provide more detail in the data synthesis section. Specifically, the authors need to consider how they will synthesise data for estimates of prevalence, what transformation they will use and if necessary/appropriate what estimator for the between-study heterogeneity will be used.

Response: We have now provide more details on the data synthesis methodology: (lines 264-267). Heterogeneity assessment methodology is described (lines 261-263).

7. What will the authors do if there aren't enough studies to undertake a meta-analysis? And how will you determine if there are an adequate number of studies?

Response: We thank the reviewer for this comment. We do not expect this to be an issue in our meta-analysis, as in preliminary searches, we have identified >100 studies evaluating the prevalence of respiratory viruses in stable COPD and/or exacerbations and >30 studies evaluating the impact of the presence of respiratory viruses on clinical outcomes.

We have defined all outcome measures in detail, and we will pool any available relevant data for every outcome. Inadequacy of data will be evaluated using GRADE methodology for evaluating the certainty in the body of evidence (imprecision domain).

8. The authors have proposed a total of 10 subgroup and sensitivity analyses. Potentially, the authors will end up with more subgroup/sensitivity analyses than included studies. For this reason, the authors should define a hierarchy of subgroup/sensitivity analyses.

Response: First of all, we will use our sensitivity analyses to validate the results of the main meta-analyses, and -therefore- we do not consider these would increase the risk of type 1 error. We do agree that the conduct of meta-regression analyses with 7 different factors might introduce a potential risk of type 1 error, especially in projects B and C, which are more exploratory. Instead of defining a hierarchy, we decided (i) to only conduct univariate and not multi-variate meta-regression analyses and (ii) to confidently assume effect modification in cases of meta-regression analyses with $p < 0.01$ (lines 287-290).

9. As well, in the subgroup and sensitivity analyses the authors describe these analyses in terms of patients (e.g. “We will analyse separately patients who were receiving/not receiving inhaled corticosteroids”), however the authors won’t necessarily have the data at the patient level but instead they will have the aggregate data presented in each study. How will the authors’ handle the situations where they can not disentangle the different subgroups of patients in the studies?

Response: We have now clarified that we will first attempt to use subgroup data, if they are available, but alternatively, we will use study-level variables for conducting the meta-regression analyses (lines 273-284). We have also re-worded the description of the sensitivity analyses, to avoid any confusion.

10. In the discussion section the authors mention 2 previous systematic reviews but they don’t cite the references and so it is impossible to know how this review differs from the previous publications and whether this is in fact adding anything more to the literature.

Response: We have now included the references to the two previous SRs (references 24 and 25).

Minor comments:

1. Abstract – I don’t understand the first sentence of the abstract.

Response: We have now re-phrased for clarity

2. Methods and analysis - change PRISMA-p to PRISMA-P (capitalize the final P) and include the appropriate reference.

Done

3. Eligibility criteria – lines 166-173 this section/explanation should be included in the methods rather than the eligibility criteria

Response: We thank the reviewer for this suggestion. However, we would prefer to keep this paragraph in the eligibility criteria. In this paragraph, we describe what types of studies are eligible and which RCT interventions we would consider eligible for inclusion in our meta-analysis.

4. Change the subheading ‘Literature searchers’ to “Systematic literature search”

Done

5. Last paragraph of the discussion the author's state "We expect that this work could pump-prime clinical and translational research" – what does 'pump-prime' mean? Please revise.

Response: We have now replaced the term "pump-prime" with the synonym, but potentially better known term "stimulate"

6. Is the author list for references 1 and 2 correct?

Corrected - thank you.

VERSION 2 – REVIEW

REVIEWER	Amalia Karahalios The University of Melbourne and Monash University, Australia
REVIEW RETURNED	06-Feb-2020

GENERAL COMMENTS	I am satisfied with the authors' responses to my comments and the corresponding changes to the manuscript. I have no further comments.
--